# A Unified Framework for Extensive-Form Game Abstraction with Bounds

**Christian Kroer**
Computer Science Department
Pittsburgh, PA 15213
ckroer@cs.cmu.edu

**Tuomas Sandholm**
Computer Science Department
Pittsburgh, PA 15213
sandholm@cs.cmu.edu

## Abstract

Abstraction has long been a key component in the practical solving of large-scale extensive-form games. Despite this, abstraction remains poorly understood. There have been some recent theoretical results but they have been confined to specific assumptions on abstraction structure and are specific to various disjoint types of abstraction, and specific solution concepts, for example, exact Nash equilibria or strategies with bounded immediate regret. In this paper we present a unified framework for analyzing abstractions that can express all types of abstractions and solution concepts used in prior papers with performance guarantees—while maintaining comparable bounds on abstraction quality. Moreover, our framework gives an exact decomposition of abstraction error in a much broader class of games, albeit only in an ex-post sense, as our results depend on the specific strategy chosen. Nonetheless, we use this ex-post decomposition along with slightly weaker assumptions than prior work to derive generalizations of prior bounds on abstraction quality. We also show, via counterexample, that such assumptions are necessary for some games. Finally, we prove the first bounds for how $\epsilon$-Nash equilibria computed in abstractions perform in the original game. This is important because often one cannot afford to compute an exact Nash equilibrium in the abstraction. All our results apply to general-sum n-player games.

## 1 Introduction

Game-theoretic equilibria have played a key role in several recent advances in the ability to construct AIs with superhuman performance in games with imperfect information [5, 9, 32]. In particular these results rely on computing an approximate *Nash equilibrium* [33] for the game at hand. In typical real-world situations these games are so large that even approximate equilibria are intractable. Instead, the dominant paradigm has been to first construct some smaller *abstraction* of the game, apply an iterative algorithm for computing a Nash equilibrium in the abstraction, and map the resulting strategy back to the full game. This approach was used in the recent *Libratus* agent, which beat four top poker pros in the game of heads-ups no-limit Texas hold'em [9] (in addition to abstraction and equilibrium approximation the agent also utilized real-time subgame solving [8] and action abstraction refinement). Abstraction has also been used in trading-agent competitions [39] and security games [1–3].

In practice, abstractions are generated heuristically with no theoretical guarantees on solution quality [4, 10, 14–16, 18–22, 24, 34, 36]. Ideally, abstraction would be lossless, such that implementing an equilibrium from the abstract game results in an equilibrium in the full game. Gilpin and Sandholm [17] study lossless abstraction techniques for a structured class of games. Unfortunately, lossless abstraction often leads to games that are still too large to solve. Thus, one must turn to lossy abstraction. However, significant abstraction *pathologies* (*nonmonotonicities*) have been shown in games which cannot exist in single-agent settings: if an abstraction is refined, the equilibrium strategy from that new abstraction can be worse in the original game than the equilibrium strategy

from a coarser abstraction [37]! Lossy abstraction remains poorly understood from a theoretical perspective. Results have been obtained only for various restricted models of abstraction. Basilico and Gatti [3] give bounds for the special game class called *patrolling security game*. Sandholm and Singh [35] provide lossy abstraction algorithms with bounds for stochastic games. Brown and Sandholm [6], Waugh et al. [38], Brown and Sandholm [7], and Čermák et al. [12] develop iterative abstraction-refinement schemes that have various forms of converge guarantees but they do not give solution-quality guarantees for the original game for strategies computed in limited-size abstractions.

Results which are for *extensive-form games (EFGs)* are most related to this work. Lanctot et al. [31] show that the *counterfactual regret minimization algorithm (CFR)* converges to an approximate NE when run on an imperfect-recall abstraction that is a *skew well-formed game* (SWF) with respect to the original game, where the error in the NE has a linear dependence on the number of information sets. Kroer and Sandholm [27] show that Nash equilibria and strategies with bounded counterfactual regret computed in *chance-relaxed SWF* (CRSWF) (a generalization of SWF that allows error in chance outcomes) are approximate NE in the original game, with a linear dependence on game-tree height. Kroer and Sandholm [25] show that NE computed in perfect-recall abstractions that satisfy conditions that are similar to those in CRSWF abstractions are approximate NE in the original game with a constant dependence on payoff error (as opposed to a linear dependence on height in Kroer and Sandholm [27] or linear dependence on information sets in Lanctot et al. [31]). Kroer and Sandholm [26] extend the results of Kroer and Sandholm [25] to continuous action spaces.

The results in the previous paragraph are all for disparate models of abstraction, a specific solution concept, or specific algorithm. Yet they share a common structure on the assumptions needed in order to obtain theoretical results. They assume that information sets (i.e., decision points) are aggregated into larger information sets. All pairs of information sets that are aggregated together are compared by defining a mapping between subtrees under the information sets. This mapping then requires that the payoffs are similar, the distribution over chance outcomes is similar, and for pairs of leaves mapped to each other, the leaves have the same sequence of information-set-action pairs leading to them in the abstraction. Having similar payoffs and chance-outcomes under aggregated information sets is natural. However, the requirement that information-set-action pairs are the same for leaf nodes mapped to each other is not satisfied by the best heuristic abstraction algorithms used in practice [10, 14, 24]. In this paper we develop an exact decomposition of the solution-quality error that does not require any such assumption. This is the first decomposition of solution-quality error resulting from abstraction. This decomposition depends on several quantities that prior results did not (owing to its more general and exact nature). We then show that by making a weaker variant of previous assumptions, our decomposition can recover all previous solution-quality bounds. We show via counterexample that there exist games where the assumption on information-set-action pairs is, in a sense, necessary in order to avoid large abstraction error that is not measurable by the type of technique presented here and in prior work.

Finally, we prove the first bounds for how $\epsilon$-Nash equilibria computed in abstractions perform in the original game. This is important because often one cannot afford to compute an exact Nash equilibrium in the abstraction. All our results apply to general-sum n-player games.

## 2   Extensive-form games (EFGs)

An *extensive-form game (EFG)* is a game tree, where each node in the tree corresponds to some history of actions taken by the players. Each node belongs to some player, and the actions available to the player at a given node are represented by the branches. Uncertainty is modeled by having a special player, *Chance*, that moves with some predefined fixed probability distribution over actions. EFGs model imperfect information by having groups of nodes in *information sets*, where an information set is a group of nodes all belonging to the same player such that the player cannot distinguish among them. In the original game that we are trying to solve, we assume *perfect recall*, which requires that no player forgets information they knew earlier in the game. This is a natural condition since you generally cannot force players to forget information, and it would not be in their interest to do so. Formally, an *extensive-form game* $\Gamma$ is a tuple $(H, Z, A, P, \pi_0, \{\mathcal{I}_i\}, \{u_i\})$. $H$ is the set of nodes in the game tree, corresponding to sequences (or histories) of actions. $H_i$ is the subset of histories belonging to Player $i$. $Z \subseteq H$ is the set of terminal histories, or *leaves*. $A$ is the set of actions in the game. $A_I$ denotes the set of actions available at nodes in information set $I$. $P$, the player function, maps each non-terminal history $h \in H \setminus Z$ to $\{0, \ldots, n\}$, representing the player whose turn it is to

move after history $h$. If $P(h) = 0$, the player is Chance. $\pi_0$ is a function that assigns to each $h \in H_0$ the probability of reaching $h$ due to Chance (i.e., assuming that both players play to reach $h$). An information set $\mathcal{I}_i$, for $i \in \{1, \ldots, n\}$, is a partition of $\{h \in H : P(h) = i\}$. The utility function $u_i$ maps $z \in Z$ to the utility obtained by player $i$ when the terminal history is reached.

A *behavioral* strategy $\sigma_i$ for a player $i$ is a probability distribution over actions at each information set in $\mathcal{I}_i$. A *strategy profile* $\sigma$ is a behavioral strategy for each player. The probability that $\sigma$ puts on $a \in A_I$ is denoted $\sigma(I, a)$. We let $\pi^\sigma(z)$ and $\pi^\sigma(I)$ denote the probability of reaching $z$ and $I$ respectively, if players choose actions according to $\sigma$. We likewise let $\pi^\sigma(z|I)$ and $\pi^\sigma(\hat{I}|I)$ denote the reach probabilities conditioned on being at information set $I$. If the probability of reaching $I$ is zero due to players excluding $i$ then we define . For a given strategy profile $\sigma$ we let $\sigma_{I \to a}$ denote the same strategy except that $\sigma_{I \to a}(I, a) = 1$.

We will often quantify statements over the set of leaves or information sets that are reachable from some given information set $I$ belonging to Player $i$, sometimes conditioned on taking a specific action $a \in A_I$. We let $\mathcal{Z}_I, \mathcal{D}_I \subset \mathcal{I}_i$ be the set of leaves and information sets reachable conditioned on being at information set $I$. We let $Z_I$ and $D_I \subset \mathcal{I}_i$ be the set of leaves and information sets that are reachable without Player $i$ taking any further actions before reaching them. We let $\mathcal{Z}_I^a, \mathcal{D}_I^a, Z_I^a$ and $D_I^a$ be defined analogously but conditioned on taking action $a \in A_I$.

As is usual we use the subscript $-i$ to denote exclusion of Player $i$, for example, $\sigma_{-i}$ is the set of behavioral strategies in $\sigma$ except for the strategy of Player $i$, and $\pi_{-i}^\sigma(z)$ is the probability of reaching leaf node $z$ disregarding actions taken by Player $i$, that is, assuming that Player $i$ plays to reach $z$.

## 3   Game abstractions

We start by giving an intuitive description of how we model abstraction. We are given some perfect-recall EFG $\Gamma$ for which we would like to compute a (possibly approximate) Nash equilibrium. Instead of solving $\Gamma$ directly, we assume that we are given some *abstraction* of $\Gamma$ called $\Gamma'$. Throughout we will assume that $\Gamma'$ is itself an EFG, though it is allowed to be imperfect recall, unlike the original game. The high-level idea is to compute some approximate solution to $\Gamma'$, and then use that approximate solution to construct a strategy for $\Gamma$. The type of approximate solution computed for $\Gamma'$ may vary. For example, computing an exact Nash equilibrium in $\Gamma'$ may be overkill, since what we ultimately care about is how strong of a strategy profile we get in the original game. This is especially true when the abstraction has imperfect recall, in which case a Nash equilibrium is NP-hard to find, and it suffices to find a strategy with low counterfactual regret at every information set. We consider several notions of solution to the abstract game.

Once we have an abstraction and a solution thereof, the primary question that we ask in this paper is whether we can construct a solution to the original game that is provably near-optimal. To answer this question we need a way to reason about the differences between the original game and the abstract game. We do this by setting up a mapping between the real game and the abstract game: every information set in the real game is assumed to map onto a specific abstract information set. The strategy that we construct for the real game is such that the distribution over actions at a given information set is constructed from the distribution over actions at the abstract information set that it maps onto. In order to analyze the quality of the obtained strategy we propose a two-step process for measuring differences between the real and abstract game: In the first step we think of the original game mapping onto an *information refinement* of the abstraction, where the refinement is the abstract game but with some abstract information sets refined into two or more new information sets. The information refinement has to be at least fine-grained enough to entail perfect recall, although it may be useful in practice to consider refinement even of perfect recall information sets. We set up measures of how different payoffs and probability distributions are in the original game versus in the refinement, where these measurements are based on how information sets and actions from the real game are mapped onto the refinement of the abstraction. In the second step, we measure the difference between the refinement and the abstract game. This is again done by measuring differences in payoffs and probability distributions, this time between each information set in the refinement and the larger abstract information set that it was refined from in the abstraction. This process is illustrated in Figure 1. Figure 2 shows an example of how that construction might be arrived at in practice, though note that our framework does not require the abstraction to be one that is arrived at via game-tree modifications like this.

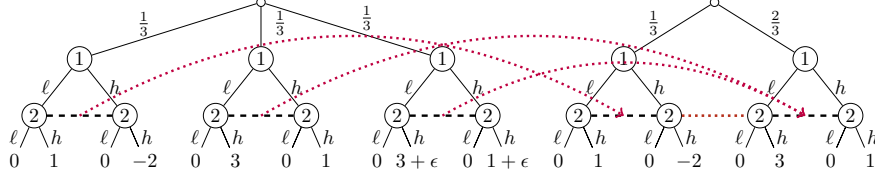

Figure 1: Abstraction example. Left: Original EFG. Right: Abstraction (which has perfect recall in this case). Dotted red arrows denote the mapping of information sets in the original game onto information set partitions in the abstract game. The dotted orange line in the abstract game denotes an information set coarsening relative to $\tilde{\mathcal{I}}$.

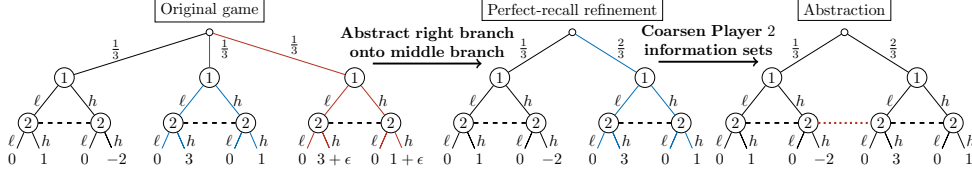

Figure 2: Example of how an abstraction could be constructed. First the rightmost red branch is removed. Second the information sets for Player 2 are coarsened as shown by the red dotted line.

The step where the original game is mapped onto a refinement would typically be used to model action removal: say we have three actions $a_1, a_2, a_3$ available at an information set, in the abstraction we may want to have only $a_1, a_2$ and consider $a_3$ as mapped onto $a_2$. The refinement step can only model information coarsening, but is very powerful for modeling certain practical types of abstraction. As an example, in poker research cards have typically been abstracted via information coarsening, say treating a pair of aces and a pair of kings as the same hand in the abstraction. We can model this in the refinement step, where aces and kings would be refined into two separate information sets.

We now give a formal description of our framework. As noted above, we consider abstractions that are themselves EFGs, but we do not require abstractions to have perfect recall (the leading practical abstractions are of imperfect recall [10, 14, 24]). We will use *the original game* to refer to some perfect-recall game $\Gamma = (H, Z, A, P, \pi_0, \{\mathcal{I}_i\}, \{u_i\})$ that we would like to compute a Nash equilibrium for. We use *the abstract game* to refer to some other game $\Gamma' = (H', Z', A', P', \pi'_0, \{\mathcal{I}'_i\}, \{u'_i\})$ that is an abstraction of $\Gamma$. The goal is to compute a (possibly approximate) equilibrium in the abstraction, and map the resulting strategy profile to the full game. For analytical purposes we will also introduce an intermediary third game, which we will refer to as the *perfect-recall refinement* $\tilde{\Gamma} = (\tilde{H}, \tilde{Z}, \tilde{A}, \tilde{P}, \tilde{\pi}_0, \{\tilde{\mathcal{I}}_i\}, \{\tilde{u}_i\})$. The perfect-recall refinement $\tilde{\Gamma}$ has the same game tree as the abstraction $\Gamma'$ (and thus $\tilde{H} = H', \tilde{Z} = Z', \tilde{A} = A', \tilde{P} = P', \tilde{\pi}_0 = \pi'_0$, and $\tilde{u} = u'$), but the information sets must be refined relative to $\Gamma$, i.e. each information set is either intact, or partitioned into several finer information sets. Thus $\tilde{\Gamma}$ has a finer-grained (i.e. less) abstraction than $\Gamma'$. $\tilde{\Gamma}$ is assumed to be a perfect-recall game, unlike $\Gamma'$. Our definition of $\tilde{\Gamma}$ is analogous to that of Lanctot et al. [31] and Kroer and Sandholm [27].

We model abstraction as a two-stage process. First, the full game is mapped onto $\tilde{\Gamma}$, with every original information set $I \in \mathcal{I}_i$ mapping onto some $\tilde{I}$ in $\tilde{\Gamma}$ via a function $f : \mathcal{I} \to \tilde{\mathcal{I}}$ that maps $\mathcal{I}$ surjectively onto $\tilde{\mathcal{I}}$. In Figure 1, each of the three original information sets belonging to Player 2 map onto the same refinement information set, but the leftmost original information set maps onto the left partition, whereas the center and right information sets map onto the right partition. In the abstract game in Figure 1, Player 2 has two subsets in $\tilde{\mathcal{I}}$: the left and right sides of their single information set. Actions are similarly mapped with an action mapping $g : A \to \tilde{A}$ that maps each $A_I$ surjectively onto $\tilde{A}_{f(I)}$. It is assumed that $f$ respects the information-set tree structure by mapping $D_I^a$ surjectively onto $\tilde{D}_{f(I)}^{g(a)}$. The final part of the first step is a way to map leaf nodes under original information sets to leaf nodes under the corresponding abstract information set. For each information set $I$ and action $a \in A_I$, we require a surjective leaf-node mapping $\psi$ from the set of leaf nodes reached below $I, a$ before player $i$ acts again, $Z_I^a$, onto $\tilde{Z}_{f(I)}^{a'}$.

The second step in our abstraction model captures the differences between the abstract game $\Gamma'$ and $\tilde{\Gamma}$. This is done by comparing the distribution over leaf nodes conditioned on being at a given

$\tilde{I} \in \tilde{\mathcal{I}}$ versus the distribution conditioned on being at the corresponding abstract information set $I'$. In Figure 1 this would correspond to comparing the leaf nodes under e.g. the right pair of nodes in Player 2's information set in the abstraction to the leaf nodes in the overall information set for Player 2. For each partition $\tilde{I}$ this is done with a set-valued map $\phi_{\tilde{I}}$ that maps the set of leaf nodes $\tilde{\mathcal{Z}}_{\tilde{I}}^{a'}$ onto $\mathcal{Z}_{I'}^{\prime,a'}$ for each $a'$ in a way such that $\{\phi_{\tilde{I}}(z') : z' \in \tilde{\mathcal{Z}}_{\tilde{I}}^{a'}\}$ specifies a partitioning of $\mathcal{Z}_{I'}^{\prime,a'}$. For a given partition $\tilde{I}$, we let $\tilde{\mathcal{D}}_{\tilde{I}}$ and $\tilde{D}_{\tilde{I}}$ be the set of descendant and child partitions, respectively, that can be reached from $\tilde{I}$.

For a strategy profile $\sigma'$ computed in $\Gamma'$ we need a way to interpret it as strategy profiles in $\Gamma$. We use the natural extension of a *lifted strategy*, originally developed by Sandholm and Singh [35] for stochastic games, to EFGs. Intuitively, a lifted strategy $\sigma^{\uparrow\sigma'}$ is a strategy where for any abstract action $a'$, the sum of probabilities in $\sigma^{\uparrow\sigma'}$ assigned to actions that map to $a'$ is equal to the probability placed on $a'$ in $\sigma'$.

**Definition 1** (Strategy lifting). *Given an abstract strategy profile $\sigma'$, a lifted strategy profile is any strategy profile $\sigma^{\uparrow\sigma'}$ such that for all $I$, all $a' \in A'_{f(I)}$: $\sum_{a \in g^{-1}(a')} \sigma^{\uparrow\sigma'}(I, a) = \sigma'(f(I), a')$.*

We use the definition of counterfactual value of an information set, introduced by Zinkevich et al. [40], to reason about the value of an information set under a given strategy profile. The counterfactual value of an information set $I$ is the expected utility of the information set, assuming that all players follow strategy profile $\sigma$, except that Player $i$ plays to reach $I$. It is defined as $V_i^\sigma(I) = \sum_{z \in \mathcal{Z}_I} \pi^\sigma(z|I) u_i(z)$ when $\pi_{-i}^\sigma(I) > 0$; otherwise it is 0. Analogously, $W_i^{\sigma'} : \mathcal{I}' \to \mathbb{R}$ is the corresponding function for the abstract game. Note that Zinkevich et al. [40] further multiply the value by the reach excluding $i$, whereas we do not. For the information set $I_r$ that contains just the root node $r$, we have $V_i^\sigma(I_r) = V_i^\sigma(r)$, which is the value of playing the game with strategy profile $\sigma$. We assume that at the root node it is not Chance's turn to move. This is without loss of generality since we can insert dummy player nodes above a root node belonging to Chance.

Kroer and Sandholm [25] showed that for an information set $I$, $V_i(I)$ can be written as a sum over descendant information sets

$$V_i^\sigma(I) = \sum_{a \in A_I} \sigma(I, a) \left[ \sum_{J \in D_I^a} \pi_{-i}^\sigma(J|I) V_i^\sigma(J) + \sum_{z \in Z_I^a} \pi_{-i}^\sigma(z|I) u_i(z) \right], \tag{1}$$

The form stated here is slightly different from the one given by Kroer and Sandholm [25]. They assume that information sets have either only leaf nodes or only information sets immediately beneath them, but this slightly more general statement follows easily from their proof. The value of $W_i(\tilde{I})$ can be written similarly.

We will show results for three different solution concepts that come up in practice. An $\epsilon$-*Nash equilibrium* is a strategy profile $\sigma$ such that $V_i^\sigma(r) \geq V_i^{\hat{\sigma}}(r) - \epsilon$ for all players $i$ and $\hat{\sigma} = (\sigma_{-i}, \hat{\sigma}_i)$. In other words, each player can gain at most $\epsilon$ by deviating to any other strategy $\hat{\sigma}_i$. This is what is computed by approaches based on first-order methods [23, 28, 29]. A *Nash equilibrium* is an $\epsilon$-Nash equilibrium where $\epsilon = 0$. Finally, a strategy profile $\sigma$ has bounded counterfactual regret if for all $i, I \in \mathcal{I}$, and $a \in A_I$, $V_i^{\sigma_{I \to a}}(I) \leq V_i^\sigma(I) + r(I)$. Strategy profiles with bounded counterfactual regret are important because regret minimization algorithms for EFGs converge by producing strategies with low $\pi_{-i}(I) r(I)$ [9, 11, 13, 30, 40].

## 4 Measuring differences between the original game and the abstract game

Our goal is to show a decomposition of the utility difference between the original game and the abstract game when using a lifted strategy. In order to do this, we need a way to measure differences between the original and the refined game. We measure payoff differences between nodes as

$$\Delta_i^R(z, \tilde{z}) = u_i(z) - \tilde{u}_i(\tilde{z}).$$

We measure leaf-node reach-probability differences conditioned on the real and abstract strategies $\sigma, \sigma'$, and action $a$, at a given information set $I$ versus its corresponding abstract information set-

partition $f(I)$ as follows

$$\Delta^P_{-i}(\tilde{z}|I, a, \sigma, \sigma') = \sum_{z \in \psi^{-1}(z'): z \in Z^a_I} \pi^\sigma_{-i}(z|I) - \pi^{\sigma'}_{-i}(z'|f(I)), \quad \text{for } z' \in Z'^{,a'}_{I'}.$$

We will also need to measure the difference in probability of reaching information set partitions, conditioned on being at the preceding information set partition belonging to the same player,

$$\Delta^P_{-i}(\tilde{I}|I, a, \sigma, \sigma') = \sum_{J \in f^{-1}(\tilde{I})} \pi^\sigma_{-i}(J|I, a) - \pi^{\sigma'}_{-i}(\tilde{I}|f(I)).$$

Note that while the set $f^{-1}(\tilde{I})$ can include information sets $J$ that do not come after $I, a$, such information sets are irrelevant since $\pi^\sigma_{-i}(J|I, a) = 0$.

We now prove a technical lemma that will be used as the primary tool for inductively proving that strategies from abstractions have bounded regret.

**Lemma 1.** *For any information set $I$ and pair of lifted strategy profiles $\sigma, \sigma'$, assume there is a bound $\Delta(J, f(J))$ such that $V^\sigma_i(J) - W^{\sigma'}_i(f(J)) \leq \Delta(J, f(J))$ for all $J \in D^a_I, a \in A_I$. Then*

$$V^\sigma_i(I) - W^{\sigma'}_i(f(I)) \leq \sum_{a \in A_I} \sigma(I, a) \Bigg[ \sum_{z \in Z^a_I} \pi^\sigma_{-i}(z|I) \Delta^R_i(z, \psi(z)) + \sum_{\tilde{z} \in \tilde{Z}^{g(a)}_{f(I)}} \Delta^P_{-i}(\tilde{z}|I, a, \sigma, \sigma') \tilde{u}_i(\tilde{z})$$

$$+ \sum_{J \in D^a_I} \pi^\sigma_{-i}(J|I) \Delta(J, f(J)) + \sum_{\tilde{I} \in \tilde{D}^{g(a)}_{f(I)}} \Delta^P_{-i}(\tilde{I}|I, a, \sigma, \sigma') W^{\sigma'}_i(\tilde{I}) \Bigg]$$

*The above holds with equality if $V^\sigma_i(J) - W^{\sigma'}_i(f(J)) = \Delta(J, f(J'))$ for all $J \in D^a_I$ and $a \in A_I$.*

We now introduce a shorthand for denoting the utility difference attributable to differences between a given information set $I$ and its abstract counterpart $f(I)$. This is the utility difference that would arise from recursively applying Lemma 1 to information sets.

$$\Delta \text{M}(I, \sigma, \sigma'_{-i}) \overset{\text{def}}{=} \sum_{a \in A_I} \sigma(I, a) \Bigg[ \sum_{z \in Z^a_I} \pi^\sigma_{-i}(z|I) \Delta^R_i(z, \psi(z)) + \sum_{\tilde{z} \in \tilde{Z}^{g(a)}_{f(I)}} \Delta^P_{-i}(\tilde{z}|I, a, \sigma, \sigma') \tilde{u}_i(\tilde{z})$$

$$+ \sum_{J \in D^a_I} \pi^\sigma_{-i}(J|I) \Delta \text{M}(J, \sigma, \sigma'_{-i}) + \sum_{\tilde{I} \in \tilde{D}^{g(a)}_{f(I)}} \Delta^P_{-i}(\tilde{I}|I, a, \sigma, \sigma') W^{\sigma'}_i(\tilde{I}) \Bigg]$$

It follows from Lemma 1 that the players' values in any lifted strategy profile in the original game are close to the players' values of the corresponding abstract strategy profile:

**Lemma 2.** *Given any abstract strategy profile $\sigma'$, any lifted strategy profile $\sigma^{\uparrow \sigma'}$ achieves utility*

$$W^{\sigma'}_i(r') = V^{\sigma^{\uparrow \sigma'}}_i(r) - \Delta \text{M}(r, \sigma^{\uparrow \sigma'}, \sigma'_{-i})$$

Next we derive an expression for the difference between an abstract information set and any $\tilde{I}$ in its partitioning. We will need a way to measure the difference between an information set $I'$ and any partition $\tilde{I}$. For reach probability, we let

$$\Delta^P(\tilde{z}|\tilde{I}, \sigma') = \pi^{\sigma'}(\tilde{z}|\tilde{I}) - \sum_{z' \in \phi_{\tilde{I}}(\tilde{z})} \pi^{\sigma'}(z'|I') \tag{2}$$

be the difference between the probability of arriving at $\tilde{z}$ conditioned on a strategy $\sigma'$ and being in partition $\tilde{I}$ of $I'$ and the probability of arriving at any leaf node $z' \in \phi^{-1}_{\tilde{I}}(\tilde{z})$ conditioned on the same strategy $\sigma'$ and being in $I'$. For reward differences we let the utility difference between a leaf node $z' \in Z_{I'}$ and its corresponding leaf node $\tilde{z} = \phi^{-1}_{\tilde{I}}(z')$ in $Z_{\tilde{I}}$ be

$$\Delta^R_i(z'|\tilde{I}) = \tilde{u}_i(\tilde{z}) - \delta_{\tilde{I}} u_i(z') \tag{3}$$

where $\delta_{\tilde{I}} > 0$ is an arbitrary scalar value that can be chosen to reflect the fact that we only need payoffs to be similar in a relative sense (for example, consider two subtrees with the same payoffs except that one subtree has *all* payoffs scaled by a constant; these subtrees are strategically equivalent).

These terms allow us to measure the difference between the value $W^{\sigma'}_i(I')$ and $W^{\sigma'}_i(\tilde{I})$ for any information set $I'$ and any $\tilde{I}$ in its partition. We let $\Delta \text{P}(\tilde{I}, \sigma')$ denote this difference.

**Lemma 3.** *For any player $i$, abstract strategy profile $\sigma'$, information set $I'$ and any $\tilde{I}$ in its partition,*
$$W_i^{\sigma'}(\tilde{I}) - \delta_{\tilde{I}}W_i^{\sigma'}(I') = \sum_{z' \in \mathcal{Z}'_{I'}} \pi^{\sigma'}(z'|I')\Delta_i^R(z'|\tilde{I}) + \sum_{\tilde{z} \in \mathcal{Z}_{\tilde{I}}} \Delta^P(\tilde{z}|\tilde{I},\sigma')u_i(z') \stackrel{\text{def}}{=} \Delta P(\tilde{I},\sigma')$$

# 5 An exact decomposition of abstraction error

Our first theorem shows that an $\epsilon$-Nash equilibrium in the abstract game maps to an $\epsilon'$-Nash equilibrium in the original game, where $\epsilon'$ depends on the difference terms introduced in the previous section. We say that the abstract game has a *cycle* if there exists a sequence of information sets $I'_1, \ldots, I'_k$ such that for all $j \neq k$ there exist nodes $h'_j \in I'_j, h'_{j+1} \in I'_{j+1}$ such that $h'_j$ is an ancestor of $h'_{j+1}$, and $I'_1$ is equal to $I'_k$. The next theorem assumes the abstract game is acyclic. This enables induction over information sets.

**Theorem 1.** *Given an $\epsilon'$-Nash equilibrium $\sigma'$ for an acyclic abstract game, any lifted strategy profile $\sigma^{\uparrow\sigma'}$ is an $\epsilon$-Nash equilibrium in the original game where $\epsilon = \max_{i \in N} \epsilon_i$ and*

$$\epsilon_i = \epsilon' + \Delta M(r, \sigma^*, \sigma'_{-i}) - \Delta M(r, \sigma^{\uparrow\sigma'}, \sigma'_{-i}) + \sum_{I \in \mathcal{I}_i} \pi^{\sigma^*}(I)\left[\Delta P(f(I), \sigma^{*'}_{I' \to I}) - \Delta P(I'_I, \sigma^{*'})\right]$$

*here $\sigma^* = (\sigma_i^*, \sigma^{\uparrow\sigma'}_{-i})$ is $\sigma^{\uparrow\sigma'}$ except Player $i$ plays any best response strategy for the original game, $\sigma^{*'} = (\sigma_i^{*'}, \sigma'_{-i})$ is such that $\sigma^{*'}(I', a') = \sum_{g^{-1}(a')} \sigma^*(I, a)$ where $I \in f^{-1}(I')$ is chosen for each $I'$ in order to maximize $W_i^{\sigma^{*'}}(r)$, and $\sigma^{*'}_{I' \to I}$ is $\sigma^{*'}$ except that at $I'$ we set the strategy according to $I$, i.e. $\sigma^{*'}(I', a') = \sum_{g^{-1}(a')} \sigma^*(I, a)$.*

This theorem is the first to show results for mapping an $\epsilon'$-Nash equilibrium in the abstract game to an $\epsilon$-Nash equilibrium in the original game. Prior results have been for abstract strategies that are either exact Nash equilibria [25] or with bounded counterfactual regret [27, 31]. That is because all prior proofs were based on applying a worst-case counterfactual regret bound as part of the inductive step (which works for exact Nash equilibrium or strategies with bounded counterfactual regret but not $\epsilon$-Nash equilibrium); our proof instead constructs an expression for $W_i^{\sigma^{*'}}(r')$ (i.e., for the value of the whole abstract game) before using the fact that $\sigma'$ is an $\epsilon$-Nash equilibrium. We next show that our framework can also measure differences for strategies with bounded counterfactual regret.

**Theorem 2.** *For an abstract strategy profile $\sigma'$ with bounded counterfactual regret $r(I')$ at every information set $I' \in \mathcal{I}'$, any lifted strategy profile $\sigma^{\uparrow\sigma'}$ is an $\epsilon$-Nash equilibrium with*

$$\epsilon = \max_{i \in N} \epsilon_i, \qquad \epsilon_i \leq \sum_{I \in \mathcal{I}_i} \pi^{\sigma^*}(I)\left[\delta_{f(I)_I}r(f(I)) + \Delta P(f(I), \sigma'_{I \to \sigma^{*'}}) - \Delta P(I'_I, \sigma')\right]$$
$$+ \Delta M(r, \sigma^*, \sigma'_{-i}) - \Delta M(r, \sigma^{\uparrow\sigma'}, \sigma'_{-i})$$

*where $\sigma^* = (\sigma_i^*, \sigma^{\uparrow\sigma'}_{-i})$ is $\sigma^{\uparrow\sigma'}$ except for Player $i$ best responding, and each $\sigma'_{I \to \sigma^*}$ is equal to $\sigma'$ except that $\sigma'_{I \to \sigma^*}(f(I), a') = \sum_{a \in g^{-1}(a')} \sigma^*(I, a)$ for all $a' \in A_{f(I)}$.*

We will show in the next sections that our two main theorems generalize prior results. In addition, our theorems are the first to give an exact expression for the abstraction error; the inequalities arise only from inexactly solving the abstract game.

# 6 Generalizing prior results

We now show that if the error in each conditional distribution over child leaves and information sets depends only on error at Chance nodes then the exact results from the previous section subsume all prior solution quality bounds for EFGs [25, 27, 31] (which also make that assumption or stronger assumptions). For that we define measures of how well Chance outcomes are approximated in the abstraction:

$$\Delta_0(h, \tilde{a}) = \sum_{a \in g^{-1}(\tilde{a})} \sigma_0(h, a) - \sigma_0(\tilde{h}, \tilde{a}), \qquad \Delta_0(h) = \sum_{\tilde{a} \in \tilde{A}_{\phi(h)}} \Delta_0(h, \tilde{a})$$

Similarly for nodes in infosets belonging to Player $i$ we have the following error

$$\Delta_0(\tilde{h}|I) = \sum_{h \in I} \pi_0(h|I) - \pi_0(\tilde{h}|f(I)).$$

In order to avoid dependence on the choice of strategy for Player $i$ our result will measure the worst-case loss over pure strategies for Player $i$. We let this set be $\chi_i$. We will use $\vec{a}$ to denote a specific pure strategy, and we let $Z_I^{\vec{a}}$ denote the set $Z_I^a$ such that $a$ is the action chosen at $I$ in $\vec{a}$, and similarly for $D_I^{\vec{a}}$. In a slight abuse of notation, we let $g(\vec{a})$ denote the pure strategy in the abstract game corresponding to $\vec{a}$ when applying $g$.

**Proposition 1.** *If an abstract strategy profile $\sigma'$ and a lifted strategy profile $\sigma^{\uparrow\sigma'}$ are such that for all $i, I \in \mathcal{I}$, $\Delta_{-i,-0}^P(\tilde{z}|I, a, \sigma, \sigma') = 0$, $\Delta_{-i,-0}^P(z|I, \sigma, \sigma') = 0$, and $\Delta_{-i,-0}^P(\tilde{I}|I, a, \sigma, \sigma') = 0$ then for all players $i$ and $\sigma = (\sigma_i, \sigma_{-i}^{\uparrow\sigma'})$ we have*

$$\Delta\mathrm{M}(r, \sigma, \sigma'_{-i}) - \Delta\mathrm{M}(r, \sigma^{\uparrow\sigma'}, \sigma'_{-i}) \leq 2 \max_{\vec{a} \in \chi_i} \sum_{I \in \mathcal{I}_i} \pi^{\sigma^{\uparrow\sigma'}}(I|\vec{a}) \Bigg[ \sum_{z \in Z_I^{\vec{a}}} \pi_{-i}^{\sigma^{\uparrow\sigma'}}(z|I) \Delta^R(z, \psi(z))$$

$$+ \sum_{\tilde{z} \in \tilde{Z}_{f(I)}^{g(\vec{a})}} \Bigg[ \Delta_0(\tilde{z}[I]|I) \pi_{-i}^{\sigma'}(\tilde{z}|\tilde{z}[I]) + \sum_{h \in I} \sum_{h_0 \in \mathcal{H}_0 : h \sqsubseteq h_0} \pi_{-i}^{\sigma^{\uparrow\sigma'}}(h_0|I) \Delta_0^A(h_0) \pi_{-i}^{\sigma'}(\tilde{z}|\tilde{h}_0, a') \tilde{u}_i(\tilde{z}) \Bigg]$$

$$+ \sum_{J \in \mathcal{D}_I^{\vec{a}}} \sum_{\tilde{h} \in f(J)} \Bigg[ \Delta_0(\tilde{h}[f(I)]|I) \pi_{-i}^{\sigma'}(\tilde{h}|\tilde{h}[f(I)])$$

$$+ \sum_{h \in I} \sum_{h_0 \in \mathcal{H}_0 : h \sqsubseteq h_0} \pi_{-i}^{\sigma^{\uparrow\sigma'}}(h_0|I) \Delta_0^A(h_0) \pi_{-i}^{\sigma'}(\tilde{h}|\tilde{h}_0, \tilde{a}) \Bigg] W_i^{\sigma'}(f()I) \Bigg] \overset{\text{def}}{=} \overline{\Delta\mathrm{M}_i}(\sigma^{\uparrow\sigma'}, \sigma')$$

We can combine Proposition 1 with Theorem 1 to get a bound that is independent of the best-response strategy:

**Corollary 1.** *If $\sigma'$ is an abstract $\epsilon'$-Nash equilibrium, satisfies the condition of Proposition 1, and $\Delta P$ is zero everywhere, then any lifted strategy profile $\sigma^{\uparrow\sigma'}$ is an $\epsilon$-Nash equilibrium where $\epsilon$ is less than $\max_{i \in N} \overline{\Delta\mathrm{M}_i}(\sigma^{\uparrow\sigma'}, \sigma') + \epsilon'$*

This bound generalizes the bound of Sandholm and Singh [35] while simultaneously tightening their bound.

The game class discussed by Kroer and Sandholm [25] is easily shown to satisfy the assumptions in Proposition 1. Thus this shows a more general bound similar to that of Kroer and Sandholm [25], where we leave in several expectations rather than taking maxima everywhere (the result by Kroer and Sandholm [25] required taking several maxima where we leave in the expectation because their proof is based on upper-bounding as part of the inductive step). Therefore, Corollary 1 yields tighter results despite also being more general.

Corollary 1 shows a result for $\epsilon$-Nash equilibrium computed in the abstraction. An analogous corollary for abstract strategies with bounded immediate regret can easily be obtained by combining Proposition 1 with Theorem 2.

We now show that, similar to mapping error, if the reach of leaf nodes in the original and abstract game are the same without considering Chance moves, we can bound partitioning error with an expression that does not depend on the best response $\sigma_i^*$ of Player $i$.

**Proposition 2.** *If $\sigma'$ is such that $\pi_{-0}^{\sigma'}(\tilde{z}|\tilde{I}, a') = \pi_{-0}^{\sigma'}(z'|I', a')$ for all $\tilde{I}, a', \tilde{z}, z' \in \phi_{\tilde{I}}(\tilde{z})$, then*

$$\Delta\mathrm{P}(\tilde{I}, \sigma'_{I \to \sigma^*}) - \Delta\mathrm{P}(\tilde{I}, \sigma') \leq 2 \max_{a' \in A_{I'}} \Bigg[ \sum_{z' \in \mathcal{Z}_{I'}^{a'}} \pi^{\sigma'}(z'|I', a') \Delta_i^R(z'|\tilde{I})$$

$$+ \sum_{\tilde{z} \in \mathcal{Z}_{\tilde{I}}^{a'}} \pi_{-0}^{\sigma'}(\tilde{z}|\tilde{I}, a') \sum_{z' \in \phi_{\tilde{I}}(\tilde{z})} \Big[ \pi_0^{\sigma'}(z'|I', a')) - \pi_0^{\sigma'}(\tilde{z}|\tilde{I}, a') \Big] \Bigg] \overset{\text{def}}{=} \overline{\Delta\mathrm{P}}(\tilde{I}, \sigma'_{I \to \sigma^*}, \sigma'), \qquad \forall \sigma'_{I \to \sigma^*}$$

This can be combined with our main theorems in order to get results for $\epsilon$-Nash equilibrium or strategies with bounded regret where the partition error does not depend on the best response.

**Corollary 2.** *If $\sigma'$ has bounded counterfactual regret $r(I')$ at every information set $I' \in \mathcal{I}'$, satisfies the condition of Proposition 2, and $\Delta M$ is zero everywhere, then any lifted strategy $\sigma^{\uparrow\sigma'}$ is an $\epsilon$-Nash equilibrium where $\epsilon = \max_{i \in N} \epsilon_i$ and $\epsilon_i \leq \sum_{I \in \mathcal{I}_i} \pi^{\sigma^*}(I) \big[ \delta_{f(I)_I} r(f(I)) + \overline{\Delta\mathrm{P}}(f(I), \sigma'_{I \to \sigma^*}, \sigma') \big]$*

Kroer and Sandholm [27] took maxima in several places where we left in the expectation: they take a maximum over the decisions of Player $i$ in $\pi^{\sigma^*}(I)$, and they maximize over the partitions in $I'$. Taking these maxima avoids dependence on $\sigma^*$. Taking these maxima could easily be done in Corollary 2 as well. Kroer and Sandholm [27] also separate the difference in conditional distribution over leaves into separate terms for Chance error that occurs before and after reaching $I'$; this potentially leads to a looser bound than ours (and never tighter since we could combine our Corollary 2 with their separation). An analogue to Corollary 2 but for $\epsilon$-Nash equilibrium can be obtained by combining Theorem 1 with Proposition 2.

## 6.1 Neccessity of distributional similarity of reach probabilities

We now show that the style of bound given by Lanctot et al. [31] as well as our corrolaries 1 and 2 cannot generalize to games where opponents do not have the same sequence of information-set-action pairs, or in our case the slightly weaker requirements in Propositions 1 and 2, for game nodes that map to each other in the abstraction. The two games that we will use as counterexamples are shown in Figure 3. From the perspective of our results, the usefulness of assuming the same sequence of information-set-action pairs is that it implies the condition used in Propositions 1 and 2; the following counterexamples thus also show that this assumption is a useful way to disallow bad abstractions such as the ones presented here (although overly restrictive from a practical perspective). Contrary to the prior results, our Theorems 1 and 2 still apply to the games below. Our two theorems would give weak bounds commensurate with the large error in the abstract equilibrium; this error is contained in the terms that depend on $\Delta^P$.

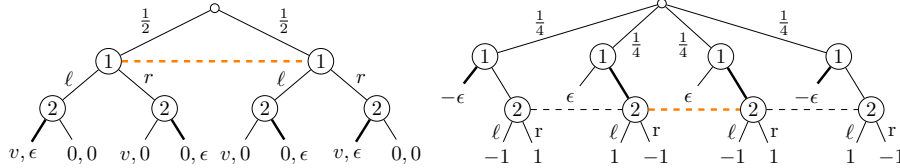

Figure 3: Left: General-sum EFG with abstraction. Right: zero-sum EFG with abstraction where Player 1 wants to minimize. Orange dashed lines denote information sets joined in the abstraction. Bold edges denote actions taken with probability 1 in the abstracted equilibrium.

On the left in Figure 3 is a general-sum game where the two nodes belonging to Player 1 are abstracted into a single information set. If we map $\ell$ onto $\ell$ and $r$ onto $r$ we get an abstraction with low payoff error: $\epsilon$ at every node. At a high level, the idea in this counterexample is that Player 2, because their nodes are not abstracted, can play opposite actions in the left and right subtrees, thus changing whether Player 1 prefers going left or right. In the original game Player 1 can react to this by choosing different actions, but not in the abstraction. Formally: Let $\epsilon > 0$. Player 2 plays the bolded edges at nodes with non-zero probability of being reached. In the abstraction, Player 1 gets $\frac{v}{2}$ for every strategy. In the full game, Player 1 can choose $\ell$ in the left subtree and $r$ in the right subtree for a payoff of $v$. Thus in every equilibrium where Player 2 plays according to the bolded edges (which includes all equilibrium refinements) Player 1 loses $\frac{v}{2}$ from abstracting, despite the payoff error being arbitrarily small. If we set $\epsilon = 0$, equilibria where Player 2 plays the bolded edges still have high loss—despite zero payoff error. This example showed that information-set-action structure has to be taken into account in order to get satisfying bounds in general. While the example is very simple (and can thus easily occur in the context of a larger game), it does exploit the fact that Player 1 utility is discontinuous in Player 2 utility. We next show that a more intricate counterexample can avoid relying on this discontinuity.

On the right in Figure 3 is a zero-sum game where the two bottom information sets belonging to Player 2 have been abstracted. Consider the following abstract equilibrium: Player 1 plays the bolded edges with probability 1, and Player 2 plays $\ell, r$ with equal probability. Player 2 gets expected utility $-\frac{\epsilon}{2}$, but in the full game Player 2 can choose $\ell$ ($r$) in the left (right) information set to get utility $\frac{1-\epsilon}{2}$. Thus Player 2 has a utility loss of $\frac{1}{2}$ despite a payoff error of $0$. The idea in this example is that, because Player 1 is not abstracted, they control the distribution over nodes in Player 2's information set in the abstraction in a way that is inconsistent with Player 2's original-game information sets: in the abstraction they get an equal distribution over nodes where $\ell$ or r is the preferred action, whereas in the original game the corresponding strategy for Player 1 means that they know exactly which node they are at.

**Acknowledgments**

This material is based on work supported by the National Science Foundation under grants IIS-1718457, IIS-1617590, and CCF-1733556, and the ARO under award W911NF-17-1- 0082. Christian Kroer is supported by a Facebook Fellowship.

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
