[Reviews · NeurIPS 2018]

Reviewer 1



This paper advances a line of work exploring how to approximate the Nash equilibrium of a game that's too large to compute directly. The idea is to create a smaller abstraction of the game by combining information sets, solve for equilibrium in the smaller game, then map the solution back to the original game. The topic relates to NIPS since this is a state-of-the-art method to program game-playing AI agents like poker bots. The authors prove new bounds on the error of the approximation that are very general. The authors provide the first general proof that an e'-Nash equilibrium in an abstraction leads to an e-Nash equilibrium in the original game. The proof requires that the abstract game is acyclic. The authors show that under a further condition that the reach of nodes is the same in the two games, then the new bounds equal or improve all prior bounds. They show by couterexample that the condition about reach is necessary for prior bounds but not for the authors' new bounds. The paper makes a solid theoretical contribution. The authors prove a general and natural approximation bound and the proof appears nontrivial. It's hard to say the practical value of the bounds. I would have liked to have seen some empirical analysis of the tightness of the bound, even on synthetic data. Does the bound help in deciding which lifted strategy profile to choose? In general, to what extent is the bound constructive so that it helps guide how to merge information sets and form abstractions? The paper appears hastily or sloppily written with repetitive phrasing and many parenthetical statements. I believe the prose could benefit from some more attention. In Section 6, what is the reach of a leaf? Is it the nodes that can reach that leaf (i.e., the leaf's "basin of attraction")? So is reach a bidirectional concept? I had thought reach was a forward concept, going from root toward leaf only. The paper is heavy with notation. I admit some of this can't be avoided but it can be hard to absorb all the notation. The paper becomes extremely dense with math starting in Section 4 with less description, explanation, and intuition than I would like. I'd also like to see more of the motivating connection to NIPS -- I believe it's there but it may not be obvious to every reader. In Theorem 1, e is used for the abstract game and e' is used for the original game. In lines 231-232, and Corollary 1, the usage is reversed. Minor comments: repetitive: specific to various specific avoid huge number of citations after one sentence & put citations in numerical order: [36, 4, 15, 16, 19, 20, 18, 21, 22, 24, 14, 10, 34]. This sentence is hard to understand because it uses two meanings of "similar" and I'm not sure the precise meaning of either: "Payoff and chance-outcome similarity is similar to what good practical abstraction algorithms seek to obtain." Update after author response: Thanks for your response. Your response and the other reviews, especially review 3 were helpful. I think I understand my confusion about reach. There is a notion of reach probability, which is the probability that a node or leaf is reached. This depends on the node's ancestors. But there seems to be another concept in the paper of "reachable sets". These are the set of nodes that can be reached from some information set. Line 104: "set of leaves or information sets that are reachable from some given information set I belonging to Player i". The variables C_I, D_I, Z_I, etc. seem to define sets of nodes that are reachable from an information set. I take this to mean descendants of the information set. By definition, the set of nodes reachable from a leaf is empty. So in Section 6, when you say "if the reach of leaf nodes and child information sets in the original and abstract game are the same", are you talking about the reach probabilities or the reachable sets? At first I assumed it was the latter, but now I think it must be the former. So do you require the reach probabilities to be exactly the same, e.g. 0.01245 for leaf L? That seems like a very strong condition. In any case, please clarify the statement to say "reach probability", not "reach". Unless I am missing something.

Reviewer 2



The paper presents a theory and formalism for new bounds between epsilon-equilibrium of abstracted games to its epsilon-equilibrium in the full game. Compared to previous work, it has slightly different assumptions. While authors claim these assumptions are more general than previous work (which is technically correct), I found that this is at the cost of having resulting bounds and theoretical results trivial and not interesting. Quality: + The submission is technically sound + Previous work is well connected - The authors overstate the strengths of their work and the result (mostly in the abstract). - “We present the first exact decomposition of abstraction error for a broad class of abstractions that encompasses abstractions used in practice.” Authors do not mention any example of how their results relate to the abstractions used in practice. Clearly any abstraction has some bounds within the theorem presented in the paper, but that does not mean this bound is meaningful nor interesting for the abstractions mentioned in practice. - My main problem with the paper is that behind all the formalism, Lemmas and Theorems, there is rather trivial result - if there is a bound on the value difference of any strategy pair (and thus strategy in abstracted game and the lifted strategy) - Lemma 1+2, we can bound epsilons for optimal strategies in the abstracted and the full game. (Theorem 1). - Most of the paper (text, theorem, lemmas etc) seem to be just straightforward machinery to prove trivial result - Missing conclusions - line 158 definition of counterfactual value seems to be missing the \sigma_{-i} term - line 181: > “conditioned on reaching a given information set I versus …. “ Should be "reaching and following action a"? - The second example in the Figure 2 is nice, but I believe the authors meant Player 2 in the “...but in the full game Player 1 can choose ` (r) in the left (right) information set” Clarity: + The high level idea is reasonably well explained + Paper uses most of the standard terminology - Most of the clarity of the idea is hidden in the formalisms Originality: + New result Significance: - The result itself does not seem interesting nor significant. *****post rebuttal I acknowledge the authors rebuttal and find it satisfactory and I am increasing score to 6. I still do not see the result very interesting or useful and think the authors heavily overstate the importance and impact of this result. I would appreciate if the authors down-stated these statements, especially related to existing abstraction techniques.

Reviewer 3



SUMMARY The paper presents a bound on the quality of the strategy for extensive-form games computed using abstraction. This bound is general enough to generalize previous works and extend beyond it. The paper is purely theoretical and does not include any experiments. QUALITY The proofs presented in the supplementary materials seem to be correct. However, they take a lot of effort to verify, because of i) the technical nature of the problem, ii) the choice of notation (see Clarity), iii) the fact that a lot of work is left to the reader (proofs of Lemma 1 and Proposition 1), iv) using mathematical objects which were not properly introduced (see Q1 and Q2). CLARITY * The paper is clearly organized and well written. The only unclear part is in Section 3, and the problem there is minor (see C1). * The paper is very technical, which is unavoidable. However, the corresponding notation is sometimes unnecessarily complicated, and the logic behind it is inconsistent (see the C) comments). If the paper aims to present a unified framework for EFG abstraction bounds, I consider this a major drawback. ORIGINALITY * The main result of the paper is an exact formula describing the error caused by working with an abstracted version of an extensive form game. This yields a unified description of the error from which two different prior results can be derived. However, the presented formula can also be applied to problems which none of the prior results was able to address (Section 7). * The authors claim that their results produce tighter bounds than the prior results. This is definitely true formally, but I do not know whether it makes a useful difference in practice. While this wouldn't be the main selling point of the paper, it would be nice to illustrate the bounds on a specific example. SIGNIFICANCE The paper addresses an important problem: abstractions are a technique which is very useful (and very often used) in practice. The paper makes a major contribution to this problem by presenting a widely applicable bound on the error caused by abstraction. DETAILED COMMENTS OF SERIOUS IMPORTANCE: Q1) The explanation of the notation around pure strategies at the beginning of Section 6 is unclear and confusing. I do not understand what is meant by \pi^{\cdot}(I|\vec a) in the RHS of the bound in the proposition. As a result, I am unable check the correctness of Proposition 1. Q2) A symbol \delta_{I'_I} suddenly appears on line 208 with no explanation or definition (and is used more times after that). It should be explicitly defined when introducing abstractions. Q3) It is not obvious how the conditional reach probabilities on line 101 are defined. Since this is central to the paper, this should be noted explicitly. How are they defined when the reach probability is zero? Q4) The definition of the counterfactual value on line 158 seems to be wrong -- shouldn't there be one term which only depends on \sigma_{-i} and another which depends on the whole \sigma? Also cf. (5) from Zinkevich et al. 2007, cited at line 155. (This seems to be a typo rather than an error.) C1) On line 121, an abstraction is explained as a two-stage process, and a high-level description of the first step is given. This step is then described in more detail -- however this description spans over 20 lines. It would be better to first give a high-level description of both steps, and only then describe these steps in more detail (or improve the clarity some other way). C2) The paper deals with an 'original' game \Gamma = (H,Z,A,P,\pi_o,{\mathcal I_i},{u_i}), it's perfect-recall abstraction (H',Z',A',P',\pi'_o,\mathcal P',{u'_i}), and a general abstraction \Gamma' = (H',Z',A',P',\pi'_o,{\mathcal I'_i},{u'_i}). I have several issues related to this choice: i) For an information set I in the original game, the authors for example the object Z_I (as derived from Z and I). If I' is an information set in the abstraction, they denote the analogous object as Z_{I'}. This seems wrong, as this object is derived from Z' and I', and should therefore be denoted as Z'_{I'}. This issue concerns most of the symbols used throughout the paper and I believe the suggested change would make the technical statement much easier to process. ii) The symbols P' (for the abstract player function) and \mathcal P' (for perf.recall inf. sets) look too similar. Consider changing one of them. iii) Most of the objects underlying the perfect-recall abstraction and the general abstraction are identical. However, the reader needs to be aware at all times whether he is currently working with an object from the p.r. abstraction or the general abstraction, as different methods apply in these two contexts. I propose using different symbols in each context but mentioning that some of them in fact correspond to the same objects. For example: "\Gamma = (H,Z,A,P,\pi_o,{\mathcal I_i},{u_i}) for the original game, \hat \Gamma = (\hat H,\hat Z,\hat A,\hat P,\hat \pi_o, {\hat \mathcal I_i},\hat {u_i}) for the perfect recall abstraction, \Gamma' = (H',Z',A',P',\pi'_o,{\mathcal I'_i},{u'_i}) for the general abstraction, where with the exception of \hat \mathcal I_i and \mathcal I', all of the corresponding \hat \cdot and (\cdot)' objects are identical." Other solutions are possible, but I believe that some change is necessary. iv) The authors consistently use u_i instead of u'_i for the abstract utility function. Either fix this or mention explicitly. C3) When using two objects of the same type (e.g. two information sets), authors consistently denote the second object with a hat (e.g. I and \hat I). This is consistent with some of the prior work, but it gets extremely impractical in the presence of indices, primes etc. (e.g. \hat I'_{\hat I} on line 195). Strongly consider using adjacent alphabet letters instead (e.g. I and J instead of I and \hat I). Also applies to terminal nodes. C4) In mapping f : I \mapsto f(I)=I'_I (line 121-123), I' is understood as constant and the subscript I as a variable. This dual usage of I is unfortunate. Also, it clashes with using I' as a generic (non p.r.) abstract information set. Throughout the paper, f(I) and I'_I is used interchangeably. Moreover, a generic p.r. abstract inf. sets are also denoted as I'_I, even in context where no original inf. set I exists. I strongly suggest only using f(I), and never I'_I, and denoting a generic p.r. abstraction inf. set as \hat I (or \tilde I, I^R, I^P, I^{PR}, or anything else which is simple and doesn't clash with I'). The second part also applies to terminal nodes (e.g. use z / \hat z / z' instead of z / \hat z' / z' as generic original / p.r. abstract / abstract terminal nodes). C5) The mappings \phi_I and \phi_{I'_I}, introduced on line 134 and 149, are confusing. I suggest two changes: First, they have very different domains, so consider using different symbols for each of them (e.g. \phi and \psi, or \phi and \hat \phi). Second, drop the subscripts (e.g. \phi instead of \phi_I) -- the condition \phi(Z_{I}^a) = Z^{g(a)}_{f(I)} from line 136 has to be explicitly stated anyway). C6) The logic behind the notation introduced on lines 104-109 is inconsistent and unintuitive. For a given information set I, the authors denote the set of terminal nodes somewhere below I as \mathcal Z_I, and the set of information sets directly below I as Z_I (without \mathcal). They also work with the set of information sets somewhere (resp. directly) below I, but denote these as \mathcal D_I and \mathcal C_I. Please either change the notation for terminal nodes to \mathcal Z_I and \mathcal Y_I, or the notation for information sets to \mathcal C_I and C_I, or choose some other consistent variant (e.g. Z_{I,a} and C_{I,a} for "somewhere below" and Z_{I,a}^{ch}, C_{I,a}^{ch} for "directly below"). C7) Explain the motivation behind the 'Z^\Rrightarrow'-like symbols (line 139). After response: I agree with reviewer #2 that the usefulness of the bounds can be better motivated, but I think the bounds give us insights into the relation of the error of the abstraction and the final strategy. Try to elaborate on the lessons learned. Q2: No, there really is an undefined \delta in equation (6) in the submission I am looking at.